# Association of High-Intensity Exercise with Renal Medullary Carcinoma in Individuals with Sickle Cell Trait: Clinical Observations and Experimental Animal Studies

**DOI:** 10.3390/cancers13236022

**Published:** 2021-11-30

**Authors:** Daniel D. Shapiro, Melinda Soeung, Luigi Perelli, Eleonora Dondossola, Devaki Shilpa Surasi, Durga N. Tripathi, Jean-Philippe Bertocchio, Federica Carbone, Michael W. Starbuck, Michael L. Van Alstine, Priya Rao, Matthew H. G. Katz, Nathan H. Parker, Amishi Y. Shah, Alessandro Carugo, Timothy P. Heffernan, Keri L. Schadler, Christopher Logothetis, Cheryl L. Walker, Christopher G. Wood, Jose A. Karam, Giulio F. Draetta, Nizar M. Tannir, Giannicola Genovese, Pavlos Msaouel

**Affiliations:** 1Department of Urology, The University of Texas MD Anderson Cancer Center, Houston, TX 77030, USA; ddshapiro@wisc.edu (D.D.S.); cgwood@mdanderson.org (C.G.W.); jakaram@mdanderson.org (J.A.K.); 2Department of Genomic Medicine, The University of Texas MD Anderson Cancer Center, Houston, TX 77030, USA; MSoeung@mdanderson.org (M.S.); GDraetta@mdanderson.org (G.F.D.); 3Department of Genitourinary Medical Oncology, The University of Texas MD Anderson Cancer Center, Houston, TX 77030, USA; LPerelli@mdanderson.org (L.P.); EDondossola@mdanderson.org (E.D.); jean-philippe.bertocchio@aphp.fr (J.-P.B.); federica.carbone@nervianoms.com (F.C.); mwstarbu@mdanderson.org (M.W.S.); AYShah@mdanderson.org (A.Y.S.); clogothe@mdanderson.org (C.L.); ntannir@mdanderson.org (N.M.T.); 4David H. Koch Center for Applied Research of Genitourinary Cancers, The University of Texas MD Anderson Cancer Center, Houston, TX 77030, USA; 5Department of Nuclear Imaging, The University of Texas MD Anderson Cancer Center, Houston, TX 77030, USA; DSSurasi@mdanderson.org; 6Center for Precision Environmental Health, Baylor College of Medicine, Houston, TX 77030, USA; Durga.Tripathi@bcm.edu (D.N.T.); Cheryl.Walker@bcm.edu (C.L.W.); 7The Ronin Project, San Mateo, CA 94402, USA; mikev@projectronin.com; 8Department of Pathology, The University of Texas MD Anderson Cancer Center, Houston, TX 77030, USA; prao@mdanderson.org; 9Department of Surgical Oncology, The University of Texas MD Anderson Cancer Center, Houston, TX 77030, USA; mhgkatz@mdanderson.org; 10Department of Behavioral Science, The University of Texas MD Anderson Cancer Center, Houston, TX 77030, USA; Nathan.Parker@moffitt.org; 11Institute for Applied Cancer Science, The University of Texas MD Anderson Cancer Center, Houston, TX 77030, USA; ACarugo@mdanderson.org (A.C.); TPHeffernan@mdanderson.org (T.P.H.); 12Translational Research to Advance Therapeutics and Innovation in Oncology (TRACTION), The University of Texas MD Anderson Cancer Center, Houston, TX 77030, USA; 13Department of Pediatrics, The University of Texas MD Anderson Cancer Center, Houston, TX 77030, USA; klschadl@mdanderson.org; 14Department of Translational Molecular Pathology, The University of Texas MD Anderson Cancer Center, Houston, TX 77030, USA

**Keywords:** renal medullary carcinoma, exercise, hypoxia, risk factor

## Abstract

**Simple Summary:**

Renal medullary carcinoma (RMC) is a rare but highly aggressive malignancy that affects individuals of African descent with sickle cell trait (SCT). The driver of RMC pathogenesis is thought to be renal medullary ischemia from red blood cell sickling in the setting of SCT. Currently, no modifiable risk factors for RMC have been identified that explain why certain individuals with SCT develop RMC. Prior studies have demonstrated that high-intensity exercise increases adverse events related to red blood cell sickling. We hypothesized that high-intensity exercise may increase the risk of RMC. We used multiple sources of evidence including retrospective and prospective review of RMC patient exercise activity, objective measurements of skeletal muscle surface area, and measurement of hypoxia levels in the renal medulla of mice with SCT following high or moderate-intensity exercise. Our results suggest that high but not moderate-intensity exercise may be associated with the development of RMC among individuals with SCT.

**Abstract:**

Renal medullary carcinoma (RMC) is a lethal malignancy affecting individuals with sickle hemoglobinopathies. Currently, no modifiable risk factors are known. We aimed to determine whether high-intensity exercise is a risk factor for RMC in individuals with sickle cell trait (SCT). We used multiple approaches to triangulate our conclusion. First, a case-control study was conducted at a single tertiary-care facility. Consecutive patients with RMC were compared to matched controls with similarly advanced genitourinary malignancies in a 1:2 ratio and compared on rates of physical activity and anthropometric measures, including skeletal muscle surface area. Next, we compared the rate of military service among our RMC patients to a similarly aged population of black individuals with SCT in the U.S. Further, we used genetically engineered mouse models of SCT to study the impact of exercise on renal medullary hypoxia. Compared with matched controls, patients with RMC reported higher physical activity and had higher skeletal muscle surface area. A higher proportion of patients with RMC reported military service than expected compared to the similarly-aged population of black individuals with SCT. When exposed to high-intensity exercise, mice with SCT demonstrated significantly higher renal medulla hypoxia compared to wild-type controls. These data suggest high-intensity exercise is the first modifiable risk factor for RMC in individuals with SCT.

## 1. Introduction

Renal medullary carcinoma (RMC) is a lethal kidney malignancy that almost exclusively afflicts young individuals of African descent with sickle hemoglobinopathies, predominantly sickle cell trait (SCT), and is associated with red blood cell (RBC) sickling in the renal medulla [1,2,3]. RMC accounts for <0.5% of all renal cell carcinomas, and the median age at diagnosis is 28 years old [3]. Males are predominantly affected over females in a 3:1 ratio. All cases of RMC are characterized by loss of *SMARCB1* expression, which is a component of the SWI/SNF chromatin remodeling complex [3]. Sickle cell trait, which is the predominant sickle hemoglobinopathy affecting patients with RMC in more than 85% of cases, is the most common sickle hemoglobinopathy with a population genotype rate 55 times more prevalent than sickle cell disease [2,3]. Sickle cells are commonly found in the renal medulla of individuals with SCT, despite the absence of RBC sickling in the peripheral blood [2,3]. Sickle cell trait is found in approximately 300 million individuals worldwide [4], and between 1/20,000 to 1/39,000 will develop RMC [5]. RMC is resistant to traditional targeted therapies used in other renal cell carcinomas (RCCs), and patients have few therapeutic options. More than 90% of patients will be diagnosed with advanced (stage III or IV) RMC, and the prognosis is poor with a median survival of only 13 months from diagnosis [2,6,7]. There is a critical need to develop new ways to screen, diagnose, and prevent RMC, informed by a better understanding of what drives its unique epidemiology.

It is unclear why only some individuals with sickle hemoglobinopathies develop RMC. All reported cases are sporadic without familial clustering or identifiable genetic risk factors other than the presence of a sickle hemoglobinopathy. Previous reports hypothesized tissue hypoxia caused by RBC sickling in the renal medulla results in RMC pathogenesis [1,3,8]. The longer length of the right renal artery can worsen hypoxia in the right medulla compared to the left, and this difference may explain the predilection of RMC for the right kidney [3]. High-intensity exercise has been associated with an increased risk of exertional rhabdomyolysis and sudden cardiac death in individuals with SCT, presumably due to tissue hypoxia from RBC sickling [9,10]. To date, however, the renal sequela of exercise in the setting of SCT have not been investigated, and most studies addressing RMC have not addressed risk factors beyond the presence of SCT [2,5]. We utilized our uniquely large dataset of patients with RMC and animal models of SCT with the primary objective of determining whether high-intensity exercise is a risk factor for RMC.

## 2. Materials and Methods

### 2.1. Patient Population

From January 2011 to February 2020, we collected clinical and demographic data from patients with pathologically confirmed RMC treated at a single institution. All histology was reviewed by a genitourinary pathology expert with over 15 years of experience, and RMC samples were confirmed to be SMARCB1 negative by immunohistochemistry. To determine the presence of high-intensity physical activity, both RMC and control patient medical records were retrospectively evaluated by two observers to reach consensus. We applied the same strategy to all patients based on the American Heart Association’s “Guide to Physical Activity Assessment” [11]. We first evaluated the four dimensions of physical activity, including the mode or type of activity (e.g., walking, cycling, weightlifting), frequency of performing the activity, duration of performing the activity, and the intensity of performing the activity. In addition, we evaluated the domains of physical activity, including occupational, domestic, transportation, and leisure time. Based on reported occupations among our RMC and control patients, we only considered occupations with significant manual labor or high-intensity exercise component as indicative of significant physical activity. These occupations included construction, professional/full time athlete, and personal exercise trainers. Domestic physical activity such as housework, yard work, or childcare were not considered indicative of significant high-intensity physical activity. Patient reported exercise including participation in sporting events, bicycling, weightlifting, running, and martial arts ≥3 times per week were considered significant physical activity. The number of hours patients participated in each physical activity per week were collected; however, this was infrequently reported among patients both in the RMC and control cohorts. These criteria were applied equally to both groups. A binary physical activity index was then created to compare the RMC and matched controls. If a patient reported at least one domain of physical activity, they were tabulated as 1 or “yes” and if no physical activity domains were reported then they were tabulated as 0 or “no”. This binary system would reduce bias when comparing the proportion of physical activity between the two cohorts.

We generated a matched control group evaluated in the same department and time period to determine whether reported exercise (as measured by the activity index) and anthropometric measurements were associated with RMC using a case-control approach. Because advanced malignancies can lead to sarcopenia and cachexia [12], we ensured that control patients had similarly staged genitourinary malignancies and were each matched with an RMC patient in a 2:1 ratio based on age and biologic sex as able.

### 2.2. Anthropometric Analysis

We objectively measured the cross-sectional area of skeletal muscle (SM) and subcutaneous adipose (SA) compartments using computed tomography (CT) imaging studies. Images were obtained and analyzed at the initial diagnosis of cancer prior to initiation of systemic therapy to avoid the changes induced by systemic therapy on either SM or SA. The SM and SA areas were measured using axial images at the midpoint of the L3 vertebral body as previously described [13]. Cross-sectional areas were standardized to the square of height in meters for each patient. Analysis was performed using SliceOMatic v5.0 (TomoVision, Magog, QC, Canada).

### 2.3. Epidemiologic Comparison

We hypothesized that active-duty military service may increase the rate of high-intensity exercise and be more prevalent in our RMC cohort. We accordingly compared the proportion of military service among our RMC cohort to that of a similarly aged (age 20–40 years old) U.S. population of black individuals with SCT [5,14,15]. Details on the population estimate can be found in the Appendix A.

### 2.4. Prospective Exercise Evaluation

After evaluating our retrospective findings, we began prospectively gathering detailed information on the exercise activity of patients with RMC seen as new patients at our institution (*N* = 7) from March 2020 to January 2021. Patients graded their exercise intensity as high, moderate, or none. High-intensity exercise was defined as achieving ≥80% maximal heart rate during each exercise session [16]. Moderate-intensity exercise was defined as exercise performed in a continuous manner at lower intensities than high-intensity exercise during each exercise session [16]. Patients also reported how much time per week was spent exercising. Lastly, we recorded the type of exercise activity and how long patients had been involved in these activities prior to the diagnosis of RMC.

### 2.5. Mouse Models

Details on the generation of genetically engineered mouse models (GEMMs) of SCT and appropriate wild-type controls are provided in the Appendix A. For exercise experiments, adult untrained wild-type (*N* = 11) and SCT (*N* = 10) mice were placed on a six-lane mechanical treadmill (Columbus Instruments) that was quickly brought up to 12–15 m/min to model acute exercise. Each session lasted 10 min. For in vivo imaging system (IVIS) studies, mice were imaged prior to exercise, immediately after exercise, and one hour after exercise. Information regarding the use of IVIS can be found in Appendix A. Two different intensities were used to measure the effect of increasing exercise intensity on renal hypoxia. As previously established [17], moderate-intensity exercise was defined as 12 m/min, which correlates with 65% to 75% VO_2_ max, i.e., a brisk walk. High-intensity exercise was defined as 15 m/min, which correlates with approximately ≥80% VO_2_ max [17].

### 2.6. Statistical Analysis

To reduce bias from inappropriate regression adjustment for colliders or mediators rather than confounders, we generated structural causal models (SCMs) to codify our pre-specified model that guided statistical analyses evaluating the relationship between reported exercise history, anthropometric measurements, and the diagnosis of RMC [18,19,20]. Multivariable unconditional and conditional logistic regression models were used to determine the association between reported exercise history or anthropometric measures with RMC diagnosis (compared with diagnosis of the matched control genitourinary malignancies) following adjustment for confounders (age, biologic sex, and race) identified by the SCMs (Appendix A). In order to determine if there was an unmeasured confounder that could “explain away” the observed association between our exposure of interest (i.e., activity index) and outcome (i.e., RMC diagnosis), we performed a robust sensitivity analysis, as described by Cinelli and Hazlett, that does not require any assumptions on the functional form of the observed association or on the distribution and number of unobserved confounders [21]. To perform this analysis, we used the package *sensemakr* for STATA. We first calculated the robustness value (RV), which provides a reference point to assess the robustness of the regression coefficient of the activity index to unobserved confounding. If a confounder’s association to the exposure and to the outcome (measured by the terms of partial *R*^2^*_exposure_* and *R*^2^*_outcome_*) are both less than the RV, then the confounder cannot “explain away” the observed association [21]. Because it is difficult to determine what an ideal RV value should be, we calculated a benchmark range based on the established covariate “race”, which is known to be strongly associated with RMC diagnosis. Analysis was performed using Stata/SE v16.1 (StataCorp, College Station, TX). Data were graphed using GraphPad. Variables were compared between RMC and matched control patients using Wilcoxon rank-sum or Fisher’s exact tests. Two-way ANOVA and Tukey multiple comparison tests were used to compare the changes in HIF1α–luciferase activity for exercise experiments.

## 3. Results

### 3.1. Retrospective and Prospective Evaluation of Physical Activity

A total of 71 patients with RMC were compared to 122 control patients (Table 1, Appendix A). We were unable to find suitable matched control patients for 4 RMC patients, and 12 patients were matched 1:1. All patients with RMC had a sickle hemoglobinopathy (predominantly SCT), the majority were male (69%) and of African descent (86%), and the primary tumor was located in the right kidney three times more frequently (76%) than the left (Table 1, Appendix A). Compared with matched controls, patients with RMC had significantly higher rates of positive activity index, reported exercise, athletic involvement, and military service (Table 1). Additionally, patients with RMC had significantly poorer ECOG performance status compared to controls (Table 1). We evaluated the albumin levels obtained at the time of diagnosis as a surrogate measure of cachexia [22,23,24]. No significant difference in albumin level was found between RMC and control patients (Table 1). SM surface area was significantly higher in patients with RMC (median 59.3 cm^2^/m^2^, IQR 50.2–69.9 cm^2^/m^2^) compared with controls (median 52.4 cm^2^/m^2^, IQR 45.0–59.1 cm^2^/m^2^) (*p* = 0.001) (Figure 1a,b). Our multivariable models adjusting for confounders identified on SCMs (race, age, and sex) demonstrated a significant relationship between physical activity index and RMC diagnosis (OR 10.4, 95% CI 4.5–23.9, *p* < 0.001), as well as between higher SM surface area and RMC diagnosis (OR 1.04, 95% CI 1.00–1.09, *p* = 0.03) (Figure 1c, Appendix A). Conditional multivariable logistic regression models again demonstrated a significant relationship between physical activity index and RMC diagnosis (OR 8.9, 95% CI 3.05–26.1, *p* < 0.001) as well as SM surface area and RMC diagnosis (OR 1.02, 95% CI 1.01–1.04, *p* = 0.02). Although it was not identified as a potential confounder (Appendix A), we performed a sensitivity analysis where albumin, a surrogate for cachexia [22], was included as a covariate in our multivariable model for SM, and SM remained significantly associated with RMC diagnosis (Appendix A). The addition of smoking as another potential confounder in our multivariable regression models did not change the significant association between both activity index and SM with RMC diagnosis (Appendix A). No significant difference was found in subcutaneous adipose surface area (*p* = 0.2), suggesting no marked difference in nutritional status between the RMC and control cohorts. Consistent with our retrospective findings, prospective evaluation of patients with RMC demonstrated that the majority of patients (71%) reported frequent, high-intensity exercise defined as ≥80% of maximal heart rate during each exercise session (Appendix A).

### 3.2. Sensitivity Analysis

Because unexplained variables may exist that could confound the relationship between activity index and RMC diagnosis, we quantified how strong such unmeasured confounders would need to be to “explain away” the association of activity index with RMC diagnosis. Our multivariable regression model demonstrated a large robustness value (RV) of 39% for the activity index. This means that confounders explaining less than 39% of the residual variance of both the exposure and outcome would not be strong enough to nullify the estimated effect of physical activity on RMC diagnosis. Race is a powerful known confounder adjusted for in our regression model. If we wanted to identify putative unmeasured confounding variables as strong as race, their total effect would be bound by the limits of partial *R^2^_exposure_* = 11% to *R^2^_outcome_* = 23%. Based on the RV for activity index, the effect of all unmeasured confounding variables would need to be over two times stronger than the effect of race in order to explain away the statistically significant association of activity index with RMC diagnosis (Appendix A).

### 3.3. Comparison of Populations with and without SCT

We validated these findings in a separate epidemiologic comparison group by looking at the proportion of all black individuals with SCT who serve in the U.S. military, hypothesizing that the intense physical activity of military duty may increase the risk of RMC. Details on the population estimate can be found in the Appendix A under the section “Epidemiologic comparison”. We found that a higher proportion (OR 8.4, 95% CI 3.2–18.4, *p* < 0.0001) of black patients with RMC (7/61, 11%) had served in the U.S. military than would be expected compared to the similarly-aged black population with SCT in the U.S. (17,816/1165, 297, 1.5%) (Figure 1d).

### 3.4. Mouse Models of Renal Medullary Hypoxia

GEMMs of SCT were generated to selectively express a TdTomato (TdT) fluorescence reporter in the kidney epithelium (Figure 2a). Pathological evaluation demonstrated RBC sickling in the renal medulla (Figure 2b) but not in the peripheral blood (Appendix A) consistent with the clinical observation that the renal medullary microenvironment is conducive to RBC sickling in individuals with SCT. To measure hypoxia in the renal medulla, we injected the Hypoxyprobe, which confirmed the presence of hypoxia in the renal inner medulla of SCT mice compared with wild-type controls (*p* < 0.0001) (Figure 2c,d). Renal hypoxia measured by the Hypoxyprobe was most pronounced in the right kidney of GEMM with SCT (Figure 2d).

To test whether increasing the intensity of exercise would worsen renal hypoxia among SCT mice, we generated a GEMM of SCT by crossing the hα/hα::β^S^/β^S^ strain with the *Gt(ROSA)26Sor^tm2(HIF1A/luc)Kael^* strain, allowing non-invasive monitoring of HIF1α activity in response to hypoxia by measuring luciferase activity using IVIS (Figure 3a) [25]. We evaluated wild-type and SCT mice at moderate and high (Figure 3b,c) treadmill exercise intensities, as previously established [17]. Luciferase fluorescence was measured before exercise, immediately after exercise, and after one hour of rest (Figure 3d,e). When exposed to moderate-intensity exercise (Figure 3d), there was a significant decrease in HIF1α activation in the SCT mice for both the left (*p* = 0.048) and right (*p* = 0.02) kidneys. There was no significant change in wild-type mice. High-intensity exercise (Figure 3e), however, resulted in significantly higher HIF1α activation among SCT mice compared to wild-type (*p* = 0.005), which was most pronounced within the right kidney (*p* = 0.04). The higher tissue hypoxia in the right kidney of SCT mice in the setting of high-intensity exercise corresponded to the predilection of the right kidney in humans towards hypoxia due to the longer length of the right renal artery [3]. As in humans, we found that the right renal artery length in the C57BL/6J mixed background mice used in our study was longer than the left (Figure 3f,g).

## 4. Discussion

Given the lethality of RMC, it is essential to identify modifiable risk factors that can inform future prevention strategies. Our clinical observations of frequently reported high-intensity exercise among our RMC patient population led us to hypothesize that high-intensity exercise may play a critical role in the pathogenesis of RMC. Our case-control analysis supports this hypothesis by demonstrating higher reported physical activity and objectively measured SM surface area among patients with RMC compared to matched controls. Additionally, we found significantly higher proportions of military service among patients with RMC compared to the general U.S. population of black SCT individuals. Our mouse models of SCT provide additional preclinical evidence of the consequence of high-intensity exercise on renal hypoxia in the setting of SCT, particularly in the right kidney, which is most frequently affected by RMC. Our experimental GEMM results suggest that moderate-intensity exercise is not harmful and may mitigate renal hypoxia, while increasing the intensity of exercise exacerbates renal hypoxia in SCT. These findings support human studies recommending low/moderate intensity exercise training to reduce complications related to sickle hemoglobinopathies [26,27].

No familial clustering of RMC has been reported to date, and a recent study of germline and somatic haplotypes in 14 unrelated patients with RMC harboring the sickle cell trait found no germline alleles or alterations in cancer predisposition genes or genes affecting kidney injury associated with RMC [28]. This suggests that environmental rather than genetic risk factors may interact with sickle hemoglobinopathies in patients with RMC. Our study is the first to suggest that increased renal hypoxia due to high-intensity exercise is a risk factor for RMC. While prior studies have evaluated the clinical characteristics, molecular landscape, and treatment outcomes of RMC, there is little evidence regarding potential modifiable risk factors for RMC [29,30,31,32,33,34,35,36]. The exact mechanism linking renal medullary hypoxia to tumorigenesis, and loss of the *SMARCB1* tumor suppressor in particular, remains to be determined. We have previously provided a testable conceptual model of how renal medullary hypoxia induced by RBC sickling can activate low-fidelity DNA repair mechanisms that can specifically induce the *SMARCB1* deletions and translocations found in RMC tissues [3,36]. The significantly increased renal medullary hypoxia found in our GEMMs in the setting of high-intensity exercise will allow more detailed investigation and refinement of the sequential molecular steps involved in RMC pathogenesis.

In the retrospective portion of our study, our matching of patients with RMC focused on ensuring similar cancer stage, age, and biologic sex between the two populations. An internal control of the utility of comparing our RMC cohort with their matched controls is that it reproduced the well-established associations between RMC and the presence of sickle hemoglobinopathies, as well as the high incidence of RMC among African Americans (Table 1) [2]. It also notably suggested that smoking is more frequently observed among patients with other genitourinary malignancies, many of which have previously established strong associations with smoking [37,38,39]. Furthermore, multivariable analyses adjusting for the racial imbalance between the RMC and matched controls confirmed the association of both increased skeletal muscle and reported exercise history with RMC diagnosis. Future efforts will focus on prospectively evaluating the type, duration, and intensity of exercise activities among patients with RMC and matched controls with SCT.

Our GEMMs provide the first evidence of the nature of this interaction by showing that only high-intensity exercise aggravates renal hypoxia in the setting of SCT. Because mouse models of RMC have not been developed to date, the exact steps of RMC pathogenesis remain to be elucidated. It is possible that the relationship between RMC and SCT is purely associative in nature and/or that hypoxic damage in the renal medulla induced by the sickling of RBCs is not the major driver of RMC pathogenesis. In that case, an alternative confounding causal mechanism would need to explain the distinct clinical features of RMC, such as the strong association with all sickle hemoglobinopathies and the distinct predilection towards the right kidney. These features are parsimoniously explained by renal medullary hypoxia due to RBC sickling being the key event of RMC pathogenesis, as represented in Appendix A [3]. This model predicts that any causes of increased renal hypoxia in the setting of SCT, such as high-intensity exercise as confirmed for the first time in our mouse models, can increase the risk of RMC. A putative alternative mechanism must also explain our clinical retrospective and prospective observations, which consistently demonstrated an association between high-intensity exercise and RMC. Even if RMC pathogenesis is not driven by increased hypoxia due to RBC sickling, the results of the present study would still serve as a body of clinical, epidemiologic, and animal modeling observations requiring explanation by proposed alternative causal theories [40]. We quantified how strong unmeasured putative confounders would need to be to remove the observed association between the activity index and RMC, using race as a reference because it is the strongest identifiable confounder between our RMC and matched cohorts. Our results suggest that, even if confounding variables two times stronger than race were missed from our regression analysis, this would still not account for the observed significant association between physical activity and RMC diagnosis (Appendix A).

From a clinical practice standpoint, our findings support the current safe exercise recommendations by the Centers for Disease Control and Prevention (CDC), the National Athletic Trainers’ Association (NATA), and the National Collegiate Athletic Association (NCAA) towards athletes with SCT, their coaches, and physicians [41,42,43,44,45]. These recommendations were generated due to the association between SCT and rhabdomyolysis or cardiovascular collapse in the setting of high-intensity exercise and endorse more moderate training regimens. Our findings provide no clinical evidence that moderate-intensity exercise is associated with RMC, and our animal models suggest that such exercise may reduce renal hypoxia in addition to the many other health benefits of exercise. Our study suggests that RMC should be high on the differential in young individuals with SCT presenting with a kidney mass and a history of high-intensity exercise or increased skeletal muscle mass on CT imaging. These associations may help inform future screening strategies for RMC by focusing on those individuals more likely to be diagnosed with this disease.

The major limitations of the clinical portion of this study are its retrospective observational nature and single center design. Additionally, given the rarity of RMC, our prospective cohort included only seven patients with exercise evaluation. Motivated by the present findings, further prospective multi-institutional evaluation of the association between exercise intensity and RMC is warranted. Another limitation is that four patients with RMC were unable to be matched. Matching patients with RMC to similarly aged patients with advanced genitourinary tumors seen in the same department and time period is challenging given the young age at which RMC affects individuals. Our inferences are strengthened by the multiple approaches we used to triangulate our findings and potentially refute our hypothesis. The retrospective evaluation of physical activity would be similarly biased in the RMC and matched controls as it was evaluated using the same approach (retrospective evaluation of the same binary activity score) over the same time period in a single department. These subjective reports were complemented by objective measures of SM surface area, which again showed significantly higher levels among patients with RMC compared to controls. This signal persisted after controlling for confounders including race, age, gender, and smoking. We additionally considered whether cachexia could have biased our findings. If that was the case, the results would have been biased against our hypothesis, because RMC is one of the deadliest genitourinary malignancies, with sarcopenia quickly manifesting following diagnosis [2,7]. This distinct aggressiveness is reflected by the significantly worse performance status we found in patients with RMC compared with matched controls. SCMs revealed performance status to be a collider (Appendix A), and this variable was omitted from multivariable regression analyses to prevent introduction of collider bias [18,19]. In addition, serum albumin, which is associated with cachexia and mortality [23,24], was similar between our RMC and matched controls and when adjusted for, did not impact the association between SM and RMC diagnosis. Furthermore, to avoid therapy effects on anthropometric measures, all imaging used to quantify muscle and adipose tissue was obtained prior to the initiation of systemic therapy. When we began to prospectively interrogate additional patients with RMC, which allowed more granular evaluation of physical activity compared with the binary activity score of our retrospective cohort, a pattern of high-intensity exercise in the majority of patients was observed. The association between RMC and high-intensity exercise was also found when we performed an epidemiologic analysis of military service.

We supplemented these clinical observations with mouse models developed to test our hypothesis that high-intensity exercise will increase renal medullary hypoxia. We found that renal medullary hypoxia significantly increased predominantly in the right kidney following high- but not moderate-intensity exercise, consistent with the distinct predilection of RMC toward the right kidney in humans [3]. The major limitation of our GEMM experiments is that they did not directly lead to RMC tumors. This is not unexpected given the rarity of RMC. If the biological context in our GEMMs parallels that of humans, we would anticipate that less than 1% of the mice with SCT would develop RMC, even after chronic exposure to high-intensity exercise.

## 5. Conclusions

In conclusion, our data suggest that high-intensity exercise may be a risk factor for developing RMC in individuals with SCT. Prospective multi-institutional studies are warranted to validate these findings and elucidate the impact of specific exercise regimens on RMC risk. These findings should be considered when counseling and monitoring individuals with SCT, particularly those occupationally engaged in high-intensity exercise.

## Figures and Tables

**Figure 1 cancers-13-06022-f001:**
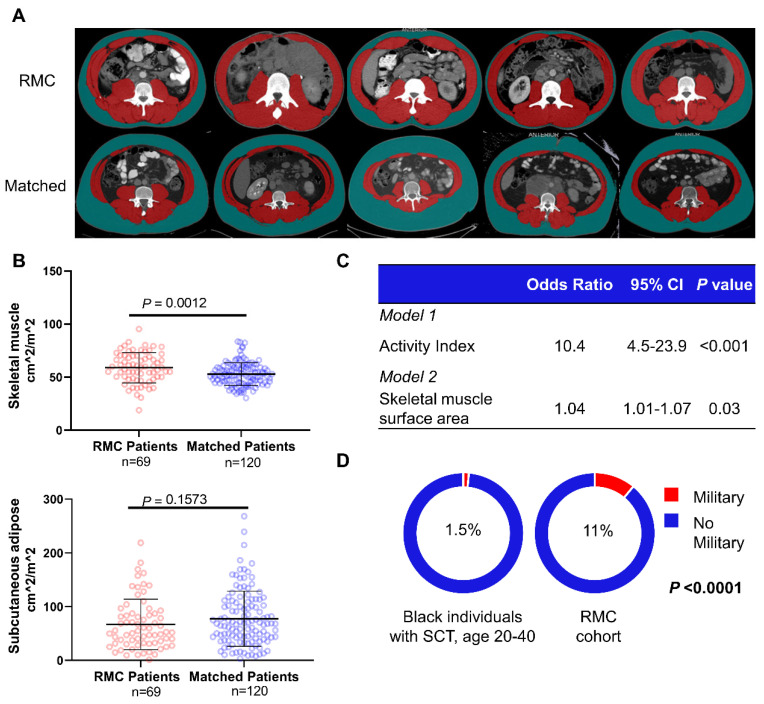
Clinical evaluation of physical activity for RMC and control patients. (**A**) Representative axial CT images from five RMC (top) and matched control (bottom) patients analyzed for skeletal muscle (SM) surface area (red) and subcutaneous adipose (green). (**B**) Dot plots of SM (top) and subcutaneous adipose (bottom) surface area. (**C**) Multivariable logistic regression models of the association between exercise history (model 1) or skeletal muscle surface area (model 2) and RMC. Additional results are provided in Appendix A. (**D**) Estimated proportion of black individuals with SCT aged 20 to 40 years in the United States military (1.5%) compared to the proportion of similarly-aged black patients with SCT in the RMC cohort (11%).

**Figure 2 cancers-13-06022-f002:**
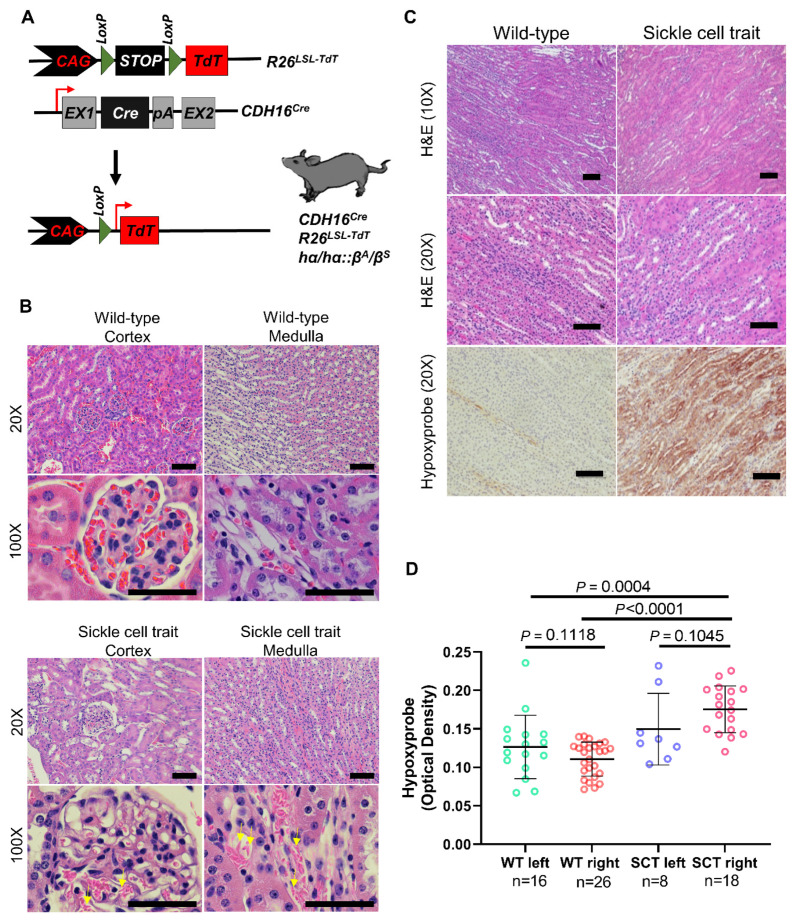
Effect of sickle cell trait on hypoxia in the renal medulla of mouse models. (**A**) Schematic of genetically engineered mouse model (GEMM) of SCT. The *CDH16^Cre^* strain was crossed with the conditional *Rosa26^LSL-TdT^* strain and the hα/hα::β^S^/β^S^ strain to generate a GEMM of SCT (hα/hα::β^A^/β^S^) that allows for tissue specific activation of TdTomato (TdT) fluorescence reporter in the kidney epithelium. (**B**) Hematoxylin and eosin staining of kidney tissue from wild-type mice (upper) and mice with SCT (lower). Sickle morphology is apparent in mice with SCT. Sickled RBCs have spindle-like morphology and are indicated with yellow arrows. (**C**) Hypoxyprobe (pimonidazole hydrochloride) was injected in the intraperitoneal cavity of GEMM of SCT with kidney-specific *CDH16^Cre^* and conditional *R26^LSL-Tom^* three hours prior to euthanasia and harvesting of kidneys. Kidneys were processed, and immunohistochemistry with the Hypoxyprobe antibody was used to show hypoxia in the inner medulla of wild-type mice (left column) and mice with SCT (right column). (**D**) Quantification of the optical density of Hypoxyprobe staining for 20x images was carried out using ImageJ. Comparisons were made between wild-type and SCT mice and stratified by kidney laterality within SCT (left kidneys, n = 8; right kidneys, n = 18) and wild-type (left kidneys, n = 16; right kidneys, n = 26) mice. Data are expressed as mean value ± SD, with *p* value calculated by student’s *t* test.

**Figure 3 cancers-13-06022-f003:**
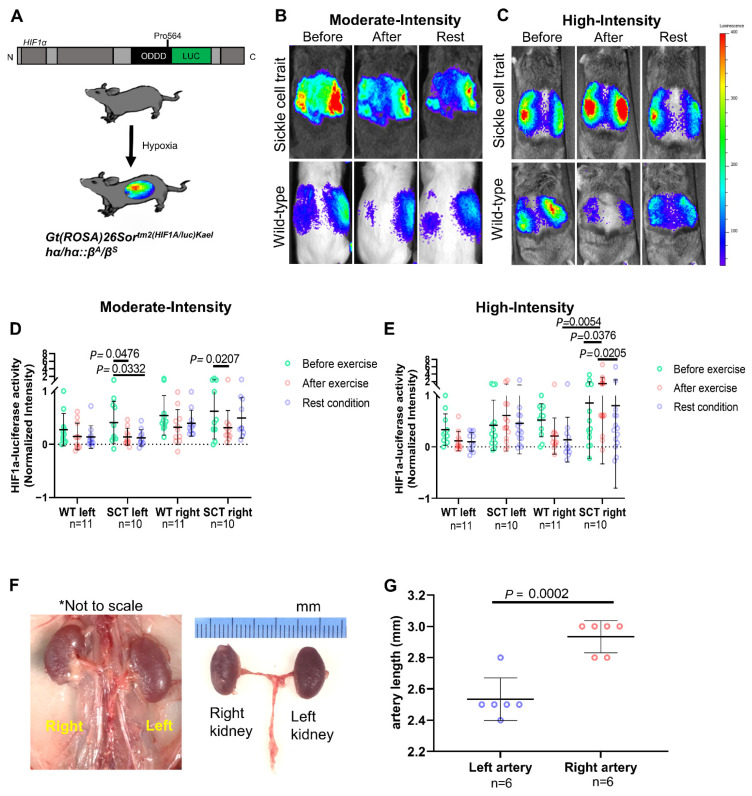
Quantification of renal hypoxia after exercise and mouse kidney anatomy. (**A**) Schematic of GEMM of SCT with HIF1α oxygen-dependent degradation domain fused to firefly luciferase, allowing non-invasive monitoring of HIF1α activity in response to hypoxia using the in vivo imaging system (IVIS). The hα/hα::β^S^/β^S^ strain was crossed with the *Gt(ROSA)26Sor^tm2(HIF1A/luc)Kael^* strain. (**B**,**C**) IVIS imaging of wild-type (n = 11) and SCT (n = 10) mice harboring the HIF1α oxygen-dependent degradation domain fused to firefly luciferase before moderate-intensity exercise (**B**) and high-intensity exercise (**C**), immediately after, and after one hour of rest. (**D**,**E**) Intensity of HIF1α-luciferase activity after moderate-intensity exercise (**D**) and high-intensity exercise (**E**). Two-way ANOVA and Tukey multiple comparison tests were used to calculate the *p* values of the changes in HIF1α–luciferase activity in adult SCT (n = 10) and wild-type (n = 11) mice after exercise and rest. (**F**) Gross anatomy of C57BL/6J mixed background mouse showing difference in the lengths between the right and left renal arteries. (**G**) Quantified difference in the length of left and right renal arteries from six adult C57BL/6J mixed background mice. Data are expressed as mean value ± SD, with *p* value calculated by student’s *t* test.

**Table 1 cancers-13-06022-t001:** Clinical characteristics of patients with RMC and matched control genitourinary malignancies.

Variable	RMC (*N* = 71)	Matched (*N* = 122)	*p* Value
Age, median (IQR)	29.6 (23.2–37.5)	29.9 (23.8–39.6)	0.5
Biological sex, no. (%)			0.7
Male	49 (69)	87 (71)	
Female	22 (31)	35 (29)	
Race, no. (%)			<0.001
African American	61 (86)	47 (39)	
White	7 (10)	66 (54)	
Hispanic	2 (3)	4 (3)	
Asian	1 (1)	5 (4)	
Sickle hemoglobinopathy, no. (%)	71 (100)	0	<0.001
Sickle cell trait	70 (99)		
Sickle beta-thalassemia	1 (1)		
Comorbidities, no. (%)			
HTN	12 (17)	33 (27)	0.2
Other comorbidities	8 (11)	58 (48)	<0.001
ECOG, no. (%)			<0.001
0	15 (21)	78 (64)	
1	42 (59)	41 (34)	
2	10 (14)	2 (2)	
3	4 (6)	1 (1)	
BMI, median (IQR), kg/m^2^	27 (24.3–29.9)	27.1 (23.9–31.6)	0.7
Albumin, median (IQR), g/dL	4.3 (4–4.6)	4.4 (4.1–4.6)	0.3
Smoking history, no. (%)	15 (21)	56 (46)	0.001
Exercise history, no. (%)	36 (51)	18 (15)	<0.001
Athlete, no. (%)	24 (34)	5 (4)	<0.001
Military, no. (%)	7 (10)	4 (3)	0.005
Activity index, no. (%)	47 (66)	20 (16)	<0.001

## Data Availability

Data may be accessed from the corresponding authors upon reasonable request.

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
