# Peer review of "Association of High-Intensity Exercise with Renal Medullary Carcinoma in Individuals with Sickle Cell Trait: Clinical Observations and Experimental Animal Studies"

_cancers, 2021, doi:10.3390/cancers13236022_

Round 1
Reviewer 1 Report
In the present study, the authors focus on renal medullary carcinoma in patients with sickle cell trait. They examine data from 71 patients with renal medullary carcinoma and matched control genitourinary malignancies, observing that high-intensity exercise might be a risk factor for renal medullary carcinoma in individuals with sickle cell trait. Apart from clinical data, the authors also investigated hypoxia levels in the renal medulla of mice with sickle cell trait after moderate- and high-intensity exercise. These mice did, however, not develop renal medullary carcinoma.
Generally, the topic of research is quite innovative and interesting. Apart from the observational study design, an important limitation is the relatively small number of patients included and animals examined. However, given the limited number of patients with renal medullary carcinoma, it is hard to acquire large datasets.
The manuscript is technically sound and well written.
Simple Summary/Abstract:
Good. However, given the relatively small number of patients included the conclusion that “high but not moderate-intensity 53 exercise is a risk factor for RMC among individuals with SCT” should be phrased with more caution.
Title: The title should include reference to the observational design of the present study. E.g., “Association of High-Intensity Exercise with Renal Medullary 2 Carcinoma in Individuals with Sickle Cell Trait: An observational study”
Introduction:
Concise and well structured.
M&M:
For limitations see comments in Discussion section.
How many years of experience did the genitourinary pathology expert have?
“To determine the presence of high-intensity physical activity, both RMC and control patient medical records were retrospectively evaluated.” – Was the retrospective evaluation of high-intensity physical activity performed by one or two observers? If two, was consensus reading performed?
Results:
Ok.
Figures:
Scatter dot plots: Please adjust the colour of the standard deviations. Due to the larger number of dots, they are difficult to read in many cases.
Discussion
Single-centre design of the present study should be discussed as a limitation.
Another limitation that needs to be discussed is the very small number of patients that whose information were prospectively collected (n=7). Finally, a very important limitation is the absence of matched controls in four patients.
Conclusion:
As the authors already report in the limitation section, the observational nature of the present study is a significant limitation. Only 7 out of the 71 patients were prospectively evaluated. Based on an observational study alone, it is very difficult to draw firm conclusions. Therefore, the conclusion should be phrased with more caution. There is certainly a need for prospective trials to validate the hypotheses phrased in the present study.
Author Response
In the present study, the authors focus on renal medullary carcinoma in patients with sickle cell trait. They examine data from 71 patients with renal medullary carcinoma and matched control genitourinary malignancies, observing that high-intensity exercise might be a risk factor for renal medullary carcinoma in individuals with sickle cell trait. Apart from clinical data, the authors also investigated hypoxia levels in the renal medulla of mice with sickle cell trait after moderate- and high-intensity exercise. These mice did, however, not develop renal medullary carcinoma.
Generally, the topic of research is quite innovative and interesting. Apart from the observational study design, an important limitation is the relatively small number of patients included and animals examined. However, given the limited number of patients with renal medullary carcinoma, it is hard to acquire large datasets.
The manuscript is technically sound and well written.
1. Simple Summary/Abstract:
Good. However, given the relatively small number of patients included the conclusion that “high but not moderate-intensity exercise is a risk factor for RMC among individuals with SCT” should be phrased with more caution.
Authors’ Reply: We have accordingly changed this to the following statement:
Line 54: “Our results suggest that high but not moderate-intensity exercise may be associated with the development of RMC among individuals with SCT.”
2. Title:
The title should include reference to the observational design of the present study. E.g., “Association of High-Intensity Exercise with Renal Medullary Carcinoma in Individuals with Sickle Cell Trait: An observational study”
Authors’ Reply: We have accordingly updated the title to accurately reflect its clinical observational component and animal experiments.
Updated Title: “Association of High-Intensity Exercise with Renal Medullary Carcinoma in Individuals with Sickle Cell Trait: Clinical Observations and Experimental Animal Studies”
Introduction:
Concise and well structured.
Authors’ Reply: Thank you for your comments.
4. M&M:
For limitations see comments in Discussion section.
How many years of experience did the genitourinary pathology expert have?
Authors’ Reply: We have added the following clarification.
Line 111: “All histology was reviewed by a genitourinary pathology expert with over 15 years of experience, and RMC samples were confirmed to be SMARCB1 negative by immunohistochemistry.”
Was the retrospective evaluation of high-intensity physical activity performed by one or two observers? If two, was consensus reading performed?
Authors’ Reply: We have added the following clarification.
Line 114: “To determine the presence of high-intensity physical activity, both RMC and control patient medical records were retrospectively evaluated by two observers to reach consensus.”
5. Results:
Ok.
6. Figures:
Scatter dot plots: Please adjust the color of the standard deviations. Due to the larger number of dots, they are difficult to read in many cases.
Authors’ Reply: We have accordingly changed the standard deviation bars to black in order to make them easier to identify.
- Discussion:
Single-centre design of the present study should be discussed as a limitation.
Authors’ Reply: We accordingly added the following statement to the limitations.
Line 425: “The major limitations of the clinical portion of this study are its retrospective observational nature and single center design.”
Another limitation that needs to be discussed is the very small number of patients that whose information were prospectively collected (n=7).
Authors’ Reply: We have now added the following statement to the limitations.
Line 426: “Additionally, given the rarity of RMC, our prospective cohort included only 7 patients with exercise evaluation. Motivated by the present findings, further prospective multi-institutional evaluation of the association between exercise intensity and RMC is warranted.”
Finally, a very important limitation is the absence of matched controls in four patients.
Authors’ Reply: We have accordingly added the following limitation.
Line 429: “Another limitation is that 4 patients with RMC were unable to be matched. Matching patients with RMC to similarly aged patients with advanced genitourinary tumors seen in the same department and time period is challenging given the young age at which RMC affects individuals.”
8. Conclusion:
As the authors already report in the limitation section, the observational nature of the present study is a significant limitation. Only 7 out of the 71 patients were prospectively evaluated. Based on an observational study alone, it is very difficult to draw firm conclusions. Therefore, the conclusion should be phrased with more caution. There is certainly a need for prospective trials to validate the hypotheses phrased in the present study.
Authors’ Reply: Thank you for your insightful and thorough review. We have accordingly rephrased our conclusion based on the reviewer’s feedback.
Line 468: “In conclusion, our data suggest that high-intensity exercise may be a risk factor for developing RMC in individuals with SCT. Prospective multi-institutional studies are warranted to validate these findings and elucidate the impact of specific exercise regimens on RMC risk.”
Reviewer 2 Report
I consider the manuscript "Association of High-Intensity Exercise with Renal Medullary 2 Carcinoma in Individuals with Sickle Cell Trait" having an interesting topic, with a clear and easy to be individualized on literature research title. The abstract is well structured.
The Introduction is too short, with few data about RMC. I recommend a larger description of RMC in this section.
Material and Methods are well conceived, to make the study reproducible, The objectives of the study should be included in Introduction Section. The Results cover all the required fields.
Discussions are well conceived and support the Results. The manuscript includes some phrases about the study limitations but the authors do not clearly mention the impact of the study on the literature research.
The Conclusions reflect the idea of the title and the manuscript presents a recent bibliography, but with a small number of titles.
Author Response
I consider the manuscript "Association of High-Intensity Exercise with Renal Medullary Carcinoma in Individuals with Sickle Cell Trait" having an interesting topic, with a clear and easy to be individualized on literature research title. The abstract is well structured.
Authors’ Reply: Thank you for your comments.
The Introduction is too short, with few data about RMC. I recommend a larger description of RMC in this section.
Authors’ Reply: We have accordingly expanded the introduction to include further description of RMC.
Line 78: “RMC accounts for <0.5% of all renal cell carcinomas and the median age at diagnosis is 28 years old [3]. Males are predominantly affected over females in a 3:1 ratio. All cases of RMC are characterized by loss of SMARCB1 expression, which is a component of the SWI/SNF chromatin remodeling complex [3]. Sickle cell trait, which is the predominant sickle hemoglobinopathy affecting patients with RMC in more than 85% of cases, is the most common sickle hemoglobinopathy with a population genotype rate 55 times more prevalent than sickle cell disease [1-4].”
Material and Methods are well conceived, to make the study reproducible.
Authors’ Reply: Thank you for your comment.
The objectives of the study should be included in Introduction Section.
Authors’ Reply: We modified the last statement of our introduction to clearly state our study’s objective.
Line 105: “We utilized our uniquely large dataset of patients with RMC and animal models of SCT with the primary objective of determining whether high-intensity exercise is a risk factor for RMC.”
The Results cover all the required fields.
Authors’ Reply: Thank you for your comment.
Discussions are well conceived and support the Results. The manuscript includes some phrases about the study limitations, but the authors do not clearly mention the impact of the study on the literature research.
Authors’ Reply: We have accordingly added to our discussion in order to address this comment.
Line 363: “Our study is the first to suggest that increased renal hypoxia due to high-intensity exercise is a risk factor for RMC. While prior studies have evaluated the clinical characteristics, molecular landscape, and treatment outcomes of RMC, there is little evidence about potential modifiable risk factors for RMC. The exact mechanism linking renal medullary hypoxia to tumorigenesis, and loss of the SMARCB1 tumor suppressor in particular, remains to be determined.”
The Conclusions reflect the idea of the title, and the manuscript presents a recent bibliography, but with a small number of titles.
Authors’ Reply: Thank you for your comment. Given the rarity of the disease, the published literature is limited. However, we have now further added to our discussion additional citations of recent related studies on RMC increasing our citations to 45 (from 37 previously). Of note, we have now included in our discussion a recent study that looked for additional genetic risk factors in patients with RMC and found no germline alleles, alterations in cancer predisposition genes or genes affecting kidney injury associated with RMC.
Line 359: “No familial clustering of RMC has been reported to date and a recent study of germline and somatic haplotypes in 14 unrelated patients with RMC harboring the sickle cell trait found no germline alleles, alterations in cancer predisposition genes or genes af-fecting kidney injury associated with RMC [28]. This suggests that environmental rather than genetic risk factors may interact with sickle hemoglobinopathies in patients with RMC.”
Reviewer 3 Report
The mamuscript shows a well designed study addressing whether High-Intensity Exercise inducing hypoxia may favor the onset of Renal Medullary
Carcinoma in Individuals with Sickle Cell Trait.
The study design is appropiate; results are well presented; statistics is correct; animal studies are interesting.
I suggest:
To better define the population uder study reported in paragraph 3.3 (result section);
To define as a limitation of the study the retrospective data from patients.
Author Response
The manuscript shows a well-designed study addressing whether high-intensity exercise inducing hypoxia may favor the onset of renal medullary carcinoma in individuals with sickle cell trait. The study design is appropriate; results are well presented; statistics is correct; animal studies are interesting.
I suggest:
To better define the population under study reported in paragraph 3.3 (result section).
Authors’ Reply: To provide further clarification regarding this population, we have now added a comment in our results section that refers to our supplementary methods which explicitly outlines the population of interest and how the population estimates were calculated.
Line 271: “Details on the population estimate can be found in the Supplementary Methods under the section “Epidemiologic comparison”.”
To define as a limitation of the study the retrospective data from patients.
Authors’ Reply: We have accordingly added the retrospective design to the limitations in the discussion section.
Line 426: “The major limitations of the clinical portion of this study are its retrospective observational nature and single center design.”
Round 2
Reviewer 1 Report
I thank the authors for their excellent revision of the manuscript. I think that it is now suited for publication in Clinical Imaging.